# Elastic–Plastic Material Deformation and Conveying Value of Twist-Free Turned Surfaces

Richard Börner [1],*, Thomas Junge [1], Thirumanikandan Subramanian [2], Stefan Thielen [2], Oliver Koch [2] and Andreas Schubert [1]

[1] Professorship Micromanufacturing Technology, Institute for Machine Tools and Production Processes, Chemnitz University of Technology, 09107 Chemnitz, Germany
[2] Institute of Machine Elements, Gears and Tribology, Technische Universität Kaiserslautern, 67653 Kaiserslautern, Germany
* Correspondence: richard.boerner@mb.tu-chemnitz.de; Tel.: +49-371-531-33821

**Abstract:** Counter-surfaces for radial shaft seals are usually finished by infeed grinding to avoid macro twist structures on the surface since they can impose a conveying action on the lubricant. This can lead to either leakage or starved lubrication and subsequent thermal damage depending on the direction of said conveying action. Turning processes can offer a more cost-effective surface finish, but conventional methods cause twist structures, which can impair the leakage prevention of the sealing system. An approach for the production of twist-free surfaces was developed based on new kinematics for turning. However, the surfaces produced with this approach using case hardened specimens made from the steel 16MnCr5 show deviating structural characteristics compared to the kinematic simulation. The causes of this and the resulting influence on the conveying value are the subjects of the research work. For this purpose, in addition to hardened steel, two other materials are considered: the steel 16MnCr5 in the unhardened hot rolled delivery condition and brass as a material with good machinability. The results clearly show that there is a deviation in the machining behavior of the steel materials compared to the kinematic surface simulations, especially in the repeatedly turned areas. This is mainly due to elastic–plastic deformation effects. Despite the actually twist-free surface profile, certain characteristics result in an anisotropic structure, which partially has an influence on the conveying value.

**Keywords:** conveying value measurements; elastic–plastic material deformation; hard machining; sealing surfaces; surface characterization; twist-free turning

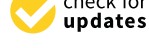



## 1. Introduction

Radial shaft seals are predominantly used for sealing rotating parts in lubricated systems. It is their purpose to prevent the leakage of lubricant and the penetration of fluids or dust from outside [1]. A total of 30% of the failures of the radial shaft seal functionality can be attributed to the shaft machining. Hence, the finish of the counter-surface of the radial shaft seal has a considerable influence on the sealing system. On the one hand, it determines the wear processes taking place at the sealing lip. On the other hand, its micro and macro structures can lead to a conveying effect in the axial direction, which is often dependent on the direction of rotation. If this occurs in the direction of the oil side, the system is tight. However, if it exceeds a limit value, this can lead to insufficient lubrication or dry running and, thus, to an increase in friction and temperature in the sealing gap. The consequences are premature wear and leakage. Hence, different processes have been applied for the manufacturing of twist-free surfaces [2].

Grinding is one of the most commonly used methods for finish machining of counter-surfaces for radial shaft seals as it allows for the generation of twist-free surfaces, which impede significant conveying during rotation [3]. Nevertheless, from an economic and environmental point of view, this process entails numerous disadvantages, such as high

investment costs, strongly limited flexibility, low predictability of the achieved surface, and poor energy efficiency [4,5].

As an alternative, hard turning offers advantages such as greater flexibility, the opportunity for complete and dry machining, and the elimination of spark-out times [4,6]. However, it generates surfaces with a twist structure that could lead to a rotation-dependent conveying effect of the sealed lubricant [7]. Although Klocke et al. [8] show that the hard-turned surfaces with a significant twist structure do not necessarily lead to an impairment of the sealing function of the radial shaft seal, twist-free radial shaft seal counter-surfaces are usually required from manufacturers of these parts [9]. Consequently, turning operations have been modified to obtain twist-free surfaces.

In grooving operations, machining is performed without axial feed. The disadvantages are the very high demands on the quality and the alignment of the cutting edge parallel to the workpiece axis [6]. In tangential turning, a wide cutting edge is used, which is inclined by a defined angle over the workpiece [4,10]. The point of contact moves along the cutting edge of the tool, and a twist-free surface is created. However, special machines with more complex kinematics and consequently high purchase costs are required.

Thielen et al. described a two-stage turning process, where the tool travels along the workpiece axis in the opposite direction of the primary feed motion, subsequently after a conventional turning operation. They found a dependency between the conveying value and the structure size in an axial direction. If the feed is higher than the contact width of the sealing lip (100 μm–150 μm), the conveying value changes with the axial position of the sealing lip. On the one hand, a reduction of the feed avoids this effect and can lead to a very low conveying value. On the other hand, smaller feeds increase the machining time, especially for the two-stage turning, where the machining time is already twice as large as for a conventional turning operation [7].

In conclusion, several efforts have been made to substitute the grinding operation for the manufacturing of counter surfaces for radial shaft seals with suitable twist-free turning operations. However, the high demand for special machines and tools, as well as long processing times, indicated the need for a new manufacturing process. Therefore, Zhang et al. introduced a turning process called "start-stop turning" (SST) for the generation of twist-free counter-surfaces in hard turning [11]. Although the surfaces produced by SST show zero twists, they are referred to as twist-free in this paper because of the theoretically non-existent conveying effect. During the manufacturing process, the tool performs short-stroke movements in the direction of the machine tool feed motion. In order to achieve a twist-free surface, the tool remains at the same position in relation to the feed direction for a period of at least one workpiece revolution ("stop phase"). Subsequently, the tool is quickly moved into the positive direction ("start phase") of the machine tool feed motion by a displacement complying with the desired groove width. These short, highly dynamic motions place high demands on the machine tool. Hence, Steinert et al. [12] integrated an additional short-stroke movement actuator as a drive unit into the lathe to reduce the moving masses for the short-stroke movement and improve the dynamical performance. Their investigations showed that the number of workpiece revolutions in the stop phase (dwell value) should be > 1 to enhance the quality of the twist-free surfaces. Nevertheless, the limited dynamic stability of the short stroke actuator only allowed a maximum jerk of $50 \text{ m/s}^3$ and a resulting feed velocity of 200 mm/min.

An improved drive unit with enhanced dynamical performance and mechanical stability was used by Žůrek et al. [13] for experiments in order to produce twist-free surfaces. However, they recognized a change in the cutting conditions during the start phase when the tool removes a part of the ridge between the feed grooves while transitioning from one groove to the next one. Further investigations showed that the grooves become partly deeper when machining with a non-integer dwell value [14]. Those surface imperfections could be manipulated by changing the stroke velocity in the start phase or the dwell value in the stop phase.

Currently, such functional surfaces are usually quantitatively characterized by means of the Mercedes Benz Norm 31007-7 (MBN). In addition, the surfaces are qualified by the conveyance value measurements. However, the surfaces produced with this new approach, usually using hardened steel, do not allow for a meaningful application of the MBN in several cases, such as the consideration of the whole circumferential surface, as there may be misinterpretations of the standard due to some structural deviations [13]. Furthermore, the resulting local change of the surface structure could have a major impact on the functional performance as a counter-surface for radial shaft seals.

Consequently, this paper aims to analyze the change in cutting conditions and its effect on the surface structure and the resulting functional behavior. It is assumed that elastic–plastic material deformations during hard machining are the cause of the structural deviations if the minimum undeformed chip thickness is not reached during the repeated turning of a groove. Therefore, various dwell values are applied for three different materials. Based on the different machining behavior and the resulting structures, an understanding of the process mechanisms can be obtained, which enables a more diverse application of the SST process. Finally, performance tests reveal the effect on the conveying value in operation as a counter-surface for radial shaft seals.

In comparison to the described modifications of twist-free turning in state of the art in science and technology, SST offers the possibility for a further increase in productivity on the basis of an extensive process understanding. That paves the way for an effective industrial application.

## 2. Materials and Methods

### 2.1. Start-Stop Turning Process

The concept of the SST process is characterized by a combination of movements alternating in the positive and negative direction of the machine tool feed motion. The application of an additional air-bearing drive unit mounted on the machine tool feed axis has proved to be advantageous in realizing these alternating movements. The integration of the short-stroke drive system into a lathe and its superimposed motions are shown in Figure 1.

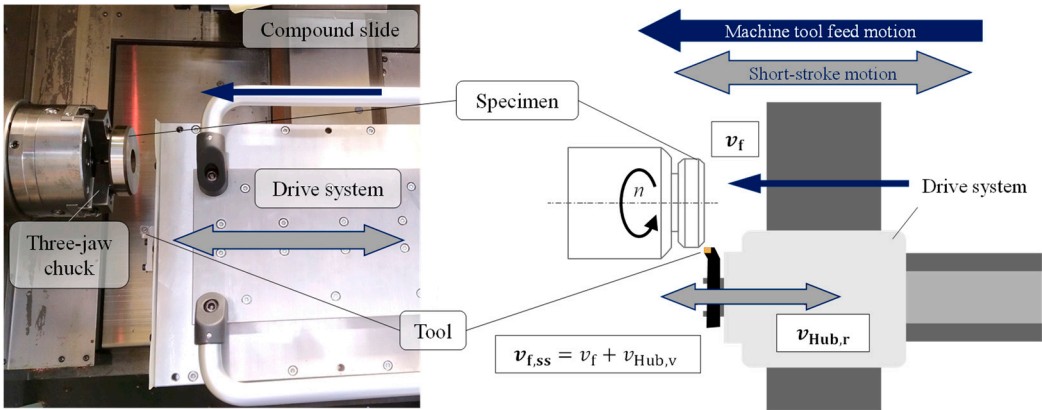

**Figure 1.** Experimental setup with air bearing drive system in SPINNER lathe (**left**) and the schematics of the motions during SST (**right**).

The mathematical description of the dynamic behavior of the SST by means of equations was first introduced in [13], and it is explained again in the following for better understanding. Because of the oscillating tool movement in the direction of feed motion, the machining process has to be separated into two different phases. As it is the main objective to machine surface structures with a zero-lead angle, the tool has to remain at the same axial position for at least one revolution of the workpiece. This is referred to as the stop phase, whereas the velocity of the tool in the negative direction of machine tool feed

motion $v_{\text{Hub,r}}$ has to comply with the machine tool Z-axis feed velocity value. This can be described by the equation

$$v_{\text{Hub,r}} = n \cdot f \,, \tag{1}$$

where $n$ is the number of revolutions of the machined workpiece per time unit and $f$ the machine feed in the direction of the Z-axis. After a defined time in the stop phase, the tool motion changes its direction and moves into the positive direction of the feed motion. This phase is referred to as the start phase. The resulting motion of the tool is depicted schematically in Figure 2 as the superposition of the feed motion of the machine tool and the short-stroke motion of the drive system.

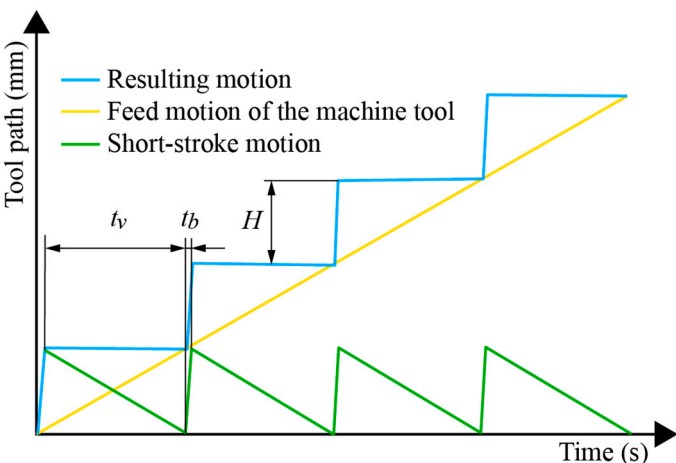

**Figure 2.** Resulting tool path during SST.

The necessary short-stroke distance $H$ can be calculated by the feed and the dwell (Equation (2)), which is characterized by the number of revolutions $n_{\text{v}}$ the tool remains in the stop phase:

$$H = f \cdot n_{\text{v}} \,. \tag{2}$$

The parameter $n_{\text{v}}$ must have at least a value of one to achieve a uniform surface structure. Otherwise, there would be no complete machining of the specimen surface.

### 2.2. Resulting Twist-Free Surface Structure

Previous investigations revealed a change in cutting conditions during the start phase when the tool removes a part of the ridge between the feed grooves while transitioning from one groove to the next one. Those effects also occurred when the tool passed over the already machined surface again. As a result, transition areas characterized by slightly deeper cavities occur. Those areas can be enlarged by a reduction of $v_{\text{Hub,v}}$ or a non-integer value of $n_{\text{v}}$ [14]. However, they affect the twist parameters that are calculated by the MBN 31007–7 in MountainsMap® 7.4 (Digitial Surf, Besançon, France) as the algorithm is not able to analyze such surface artifacts. Hence, they could also influence the functional properties as counter-surfaces for radial shaft seals. As a consequence, it is recommended to accelerate the tool as fast as possible in the start phase to reduce the impact of those transition areas. The resulting feed velocity of the tool relative to the workpiece $v_{\text{f,ss}}$ is the sum of the feed velocity $v_{\text{f}}$ and the stroke velocity $v_{\text{Hub,v}}$ shown by the equation:

$$v_{\text{f,ss}} = v_{\text{f}} + v_{\text{Hub,v}} \,. \tag{3}$$

Due to the superposition of motions, the periodical surface structure width does not correspond to the feed as in conventional turning operations. The structures of twist-free surfaces with zero lead are characterized by a groove pattern with a periodical width $DP_{\text{theor}}$, which corresponds to the path of the tool in the positive direction of the feed

motion. It can be calculated by the sum of the stroke distance H and the path that the tool travels during the start phase.

$$DP_{\text{theor}} = H + t_{\text{b}} \cdot v_{\text{f}} = f \cdot n_{\text{v}} + t_{\text{b}} \cdot v_{\text{f}} \,. \tag{4}$$

### 2.3. Kinematic Simulation

The introduced twist-free surfaces can be characterized by the tool geometry and the kinematical process parameters of the lathe and the short-stroke drive system. A simulation model, developed in the software MATLAB R2019a (The MathWorks, Inc., Natick, MA, USA) and firstly introduced for vibration-imposed milling in [15,16], was extended for the prediction and evaluation of the kinematic surface structure and transition areas during SST. The kinematical approach subtracts the tool geometry along the tool trajectory with the workpiece surface, but it does not consider elastic–plastic effects such as burr formation or thermal expansion. Therefore, the model incorporates the workpiece and the tool as dexel objects represented by discrete points ("dexels") aligned over a grid in the *x-y* plane. In order to minimize the computational effort, the tool can be described by a contour line along the corner geometry. Previous investigations revealed that the same simulation results could be obtained with such a simplified representation as long as a geometrical intersection of the flank face can be excluded [16]. Therefore, the geometrical extension of the intersecting tool body in the direction of the *x*-($\Delta x$) and *y*-axis ($\Delta y$) (Figure 3c) needs to be considered. In this context, the *x*-axis represents the direction of feed motion, and the *y*-axis the direction of primary motion.

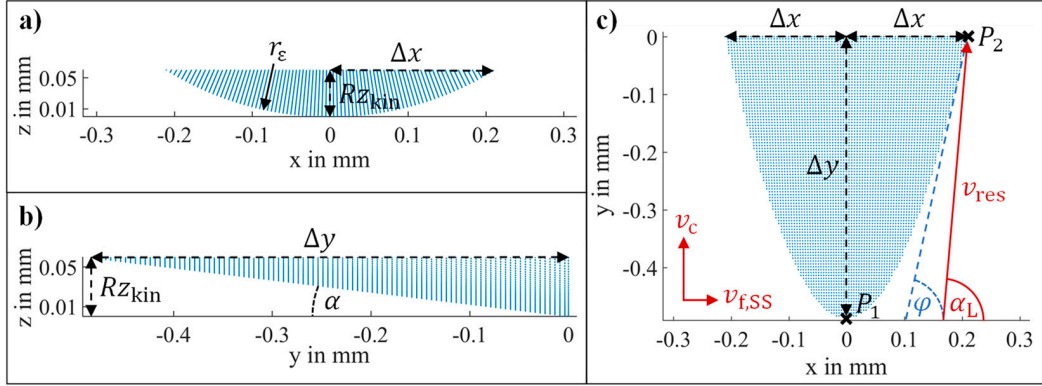

**Figure 3.** Discrete point representation of the tool body and its relevant geometrical parameters in the *x-z* plane (**a**), *y-z* plane (**b**) and *x-y* plane (**c**).

A standard tool contour (Figure 3a) in the *x-z* plane can be described by a circular equation $z(x)$. By the conversion of the formula to *x* and the substitution of *z* by the kinematical roughness height $Rz_{\text{kin}}$ Equation (5) is obtained. It represents half of the engaged tool width $\Delta x$.

$$z(x) = -\sqrt{r_\varepsilon^2 - x^2} + r_\varepsilon \;\rightarrow\; \Delta x = \sqrt{Rz_{\text{kin}} \cdot (2 \cdot r_\varepsilon - Rz_{\text{kin}})} \,. \tag{5}$$

In contrast, the equation for $\Delta y$ accounts for the influence of the clearance angle (Figure 3b):

$$\Delta y = \frac{Rz_{\text{kin}}}{\tan \alpha} = \frac{DP_{\text{theor}}^2}{8 \cdot r_\varepsilon \cdot \tan \alpha} \,. \tag{6}$$

Including the points $P_1$ ($0/\Delta y$) and $P_2$ ($\Delta x/0$) the contour line of the intersecting tool body in the *x-y* plane (Figure 3c) can be described by the quadratic equation:

$$y(x) = \frac{\Delta y}{\Delta x^2} \cdot x^2 - \Delta y \,. \tag{7}$$

The inclination angle $\varphi$ of Equation (8) at point P$_2$ is the critical angle in comparison to the transition angle $\alpha_L$. $\alpha_L$ describes the moving direction of the tool during the start phase that results from the superposition of the cutting speed $v_c$ and the resulting feed velocity $v_{f,SS}$ (Figure 3c). Thus, it represents the angular alignment of the transition area when the tool transitions from one groove to the next one:

$$\alpha_{\mathrm{L}} = arctan\left(\frac{v_{\mathrm{c}}}{v_{\mathrm{f}} + v_{\mathrm{Hub,v}}}\right) > \varphi = arctan\left(2 \cdot \frac{\Delta y}{\Delta x}\right). \tag{8}$$

As long as $\varphi$ remains smaller than $\alpha_L$, the tool flank face does not intersect with the kinematic surface structure in the transition phase. The transition angle of the tool trajectory is relatively high because $v_c$ is significantly larger than the sum of $v_f$ and $v_{Hub,v}$. Thus, the intersection of the flank face can be excluded, which allows for a simplified two-dimensional tool representation in the simulative approach. Figure 4 shows a simulated kinematical surface structure and its transition area. The inverted height scale is considered to be more informative since the depth differences of the following real surface structures, compared to Figure 5, are more clearly perceived by this display method. It is noticeable that all minimum points of the valleys and the transition area are on the same depth level.

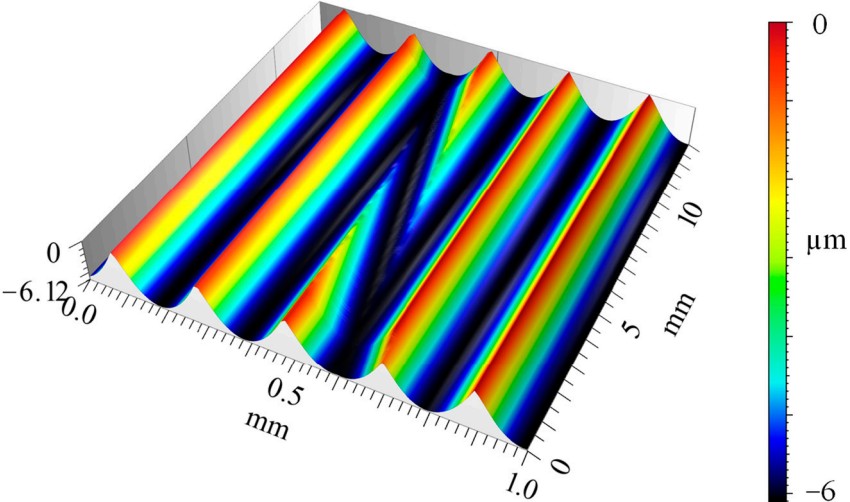

**Figure 4.** Kinematically simulated transition area.

### 2.4. Experimental Procedure

The machining experiments were carried out on a precision lathe SPINNER PD 32 (SPINNER Werkzeugmaschinenfabrik GmbH, Sauerlach, Germany). In order to realize the twist-free surfaces, an air-bearing short-stroke drive system of the company AeroLas (AeroLas GmbH, Unterhaching, Germany) was used. This has already been described in detail in [13]. The experimental setup is shown in Figure 1. In addition to the test specimens made of alloyed steel 16MnCr5 in the case hardened ((55 + 2) HRC, indentation hardness approx. 1 mm) heat treatment (H) condition and in the as-delivered condition (A—hot rolled), test specimens made of brass (CuZn39Pb3) were processed. In each case, indexable inserts with almost the same geometry and typical cutting materials were selected for machining:

- Single crystalline diamond (SCD) for machining of brass,
- Cubic boron nitride (CBN) for machining hardened steel,
- Coated cemented carbide (CCC) for machining of unhardened steel.

In addition a reference specimen was made from hardened steel without using the start-stop unit for the comparison of the resulting surface parameters as well as the effects on the conveying value. For this specimen, a twist lead angle $D_g$ of 0.0228° was measured.

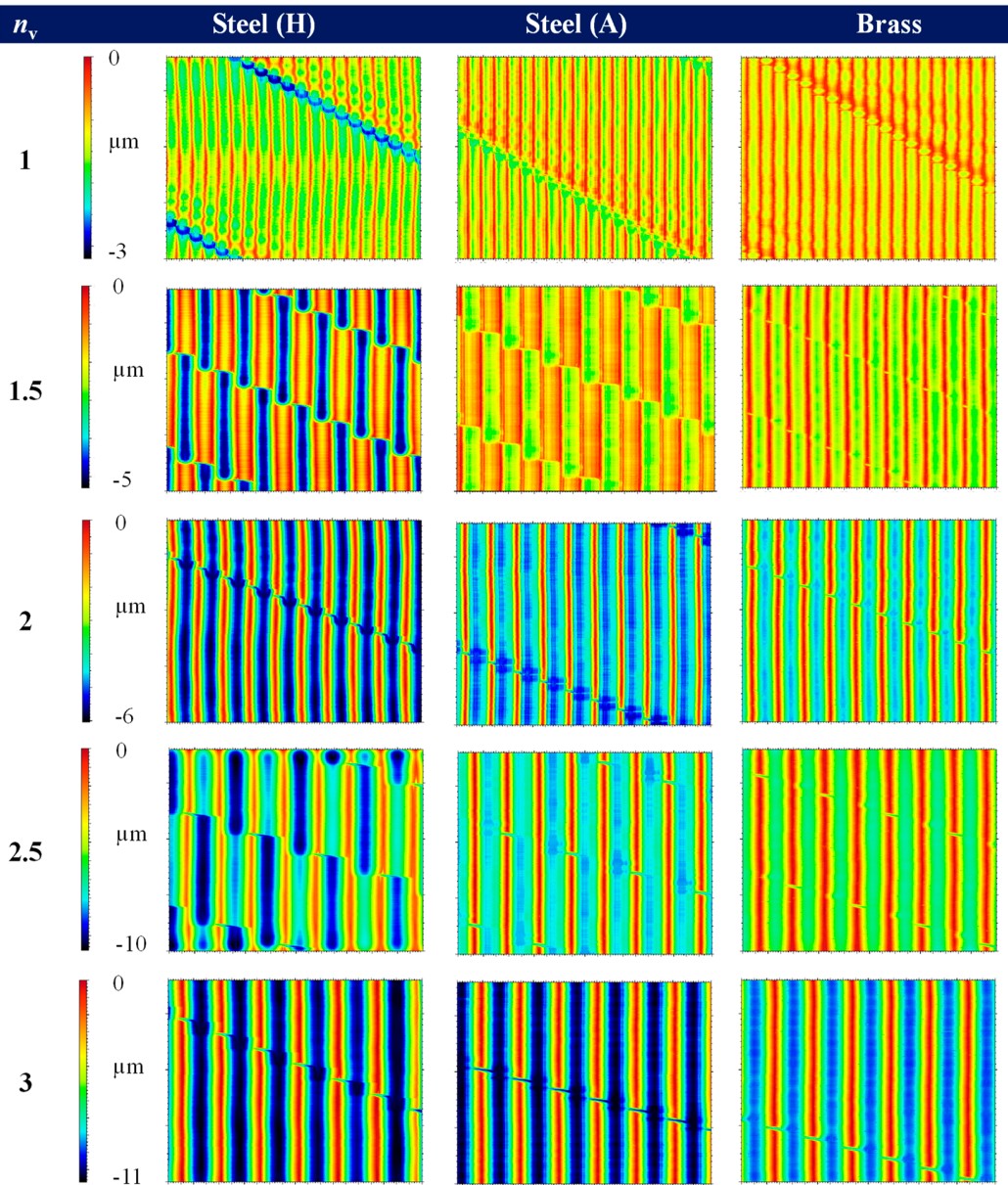

**Figure 5.** Overview of the twist-free surfaces created by SST; each field is 1.8 mm in width and 360° in length.

Indexable inserts of the uniform type CCGW 09T308 were used as tools, which had a corner radius $r_\varepsilon$ of 0.8 mm and a clearance angle $\alpha$ of 7°. The cutting edge tool angle of the insert holder in the start-stop unit was 95°. The 3D data of the cutting tools were gathered using an optical coordinate measuring machine Bruker alicona μCMM (Alicona Imaging GmbH, Graz, Austria) with a 20× magnification lens and for comparison by an optical laser scanning microscope of the type Keyence VK-9700 (Keyence Corporation, Osaka, Japan) with a 50× magnification lens. The comparison of measuring devices was used in particular because the SCD tool does not allow for a reliable characterization of the cutting edge: since, in contrast to laser scanning, the entire measuring field is illuminated, the geometrical analysis of transparent cutting material is not possible. If the tool is sputtered for better optical detection, with gold, for example, the roughness is too high compared to the cutting-edge radius. In addition, the VK-Analyzer software (Keyence) was used to determine the cutting-edge radius $r_b$. For this purpose, 200 individual profiles were created along the cutting edge in the area of the corner radius, nearly analogous to the internal measuring software of the μCMM. Since the settings of both measurements methods are

partly automated and can therefore be only limited adjusted, they differ slightly: the measuring field length of the μCMM was 270 μm, which results in a profile distance of 1.35 μm, while 250 μm (1.25 μm profile distance, respectively) was possible in the VK-Analyzer. Afterward, a radius was manually fitted in the averaged profile using 3-point measurement. For a qualitative visualization, images were taken by a scanning electron microscope (SEM) EVO MA25 (Carl Zeiss AG, Oberkochen, Germany). The results are illustrated in Table 1. They clearly show the different size ranges of the cutting-edge radius, even if the values recorded by the different measuring devices vary. Due to the previously mentioned disturbing influences, the cutting-edge radius for SCD detected by optical coordinate measurement is clearly too large. Typically, cutting-edge radii for SCD tools are $\leq 1$ μm. The measurement error in the evaluation of the laser scanning micrographs results from the parallel and not circularly arranged single profiles along the cutting edge. Note that the respective magnification was chosen in relation to relevant or distinctive features of the tools.

**Table 1.** Comparison of the used cutting tools in the unworn condition with regard to the cutting-edge radius.

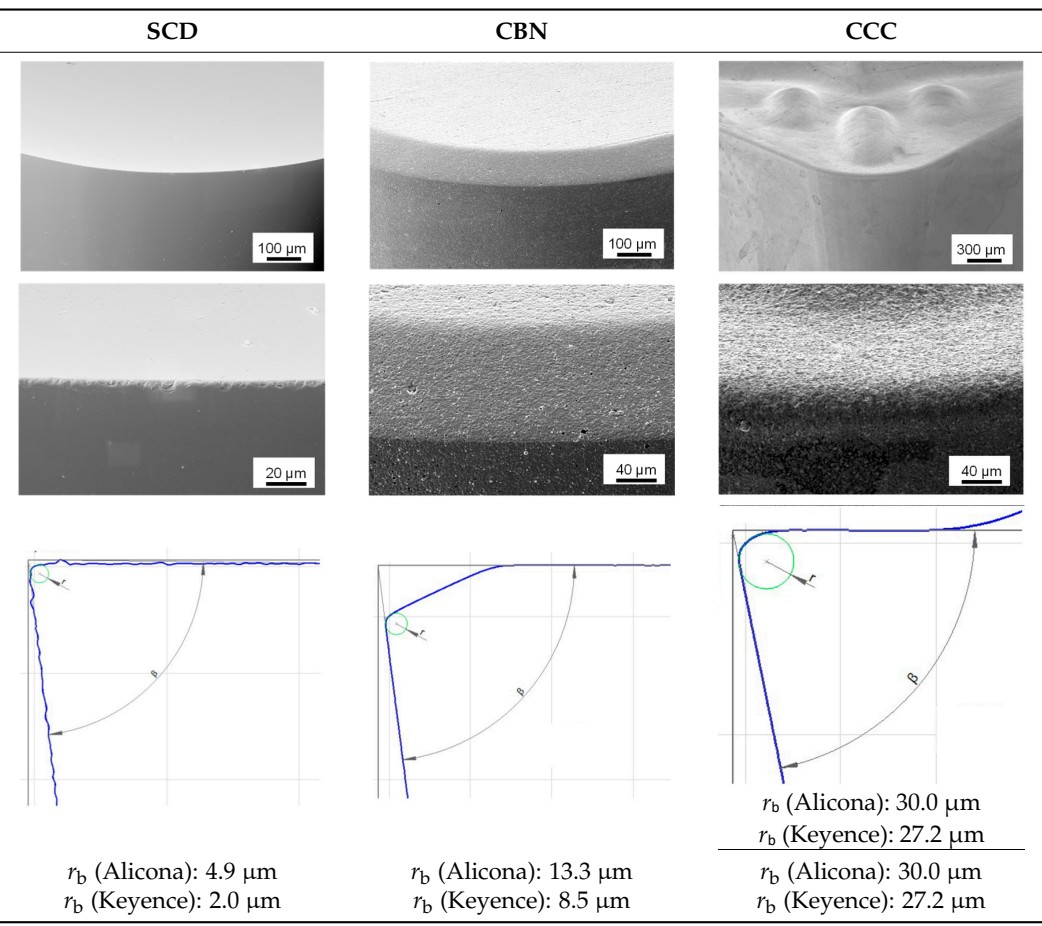

| SCD | CBN | CCC |
|---|---|---|
| $r_b$ (Alicona): 4.9 μm | $r_b$ (Alicona): 13.3 μm | $r_b$ (Alicona): 30.0 μm |
| $r_b$ (Keyence): 2.0 μm | $r_b$ (Keyence): 8.5 μm | $r_b$ (Keyence): 27.2 μm |

The 2D and 3D data of the outer surfaces were all taken tactilely by a roughness measurement system Mahr LD 120 (Mahr GmbH, Göttingen, Germany) and the shape and position measurement system Mahr MarForm MMQ 200. Furthermore, the surface analysis was carried out by the surface analysis and evaluation software MountainsMap® 7.4, which was used on the one hand for the determination of the parameters according to MBN and on the other hand for a detailed qualitative analysis of the transition areas. In order to increase the comprehensibility of the generated surface data, the individual detection methods, including possible weaknesses, are presented in the following.

As already mentioned in [13], the MBN is not suitable for a complete circumferential surface evaluation of a cylindrical specimen that is characterized by a twist-free surface structure with transition areas. However, since the standard requires either 72 individual sections per circumference or 72 individual sections distributed over 36°, the latter variant was chosen to characterize the twist-free surfaces. For a holistic representation, the specimen surfaces were therefore recorded with a resolution of 0.5°, i.e., 720 individual sections, by the position measurement system Mahr MarForm MMQ 200. On width of 10 mm, surface recording was performed for a qualitative comparison of the surface structures. For the individual profiles, the micro-roughness was filtered (cut-off 2.5 μm), and then the profiles were assembled into a surface, the shape was removed with a polynomial, and the intermediate areas were interpolated. This results in some kind of display error since this profile distance allows only limited accuracy. All surface artifacts smaller than the profile distance of 0.5°, respectively 0.35 mm, are accordingly not detected. A further measurement uncertainty results from the diameter of the stylus (diamond tip with included angle of 90° and radius of 5 μm). However, this error is sufficiently irrelevant, since the structural height differences have lateral characteristics of several millimeters. This is the reason for the use of tactile measuring technology because only in this way large-area structural height differences can be detected and thus recognized. Optical measurement methods, such as laser scanning microscopy, are not suitable for the large-area detection of curved surfaces. A corresponding section was selected in each case to realize the quantitative evaluation according to MBN. For this purpose, areas with a measurement section length of 2 mm were selected, which do not show any transition areas. Furthermore, areas of the surface showing transition areas were recorded with high resolution. For this purpose, a surface section with a width of 30° was selected for a qualitative comparison and recorded with a profile distance of 0.01° (corresponding to 3000 single scanning sections).

Additionally, the geometrical properties of the machined surfaces were measured using a stylus instrument type Mahr LD 120. These roughness measurements were carried out axially at three positions shifted by 120° on the surface with settings according to ISO 4288. The roughness parameters were determined according to ISO 4287 by applying the following specifications: a measuring distance of 4 mm and a Gaussian profile filter (ls) of 2.5 μm. The stylus was characterized by a radius of 2 μm and an included angle of 90°. According to the current calibration protocol, the measurement uncertainty is 5%, with the background noise from the ambient conditions causing an $Rz_0$ of about 11 nm.

Standardized specimens with an outer diameter of 80 mm and a width of 18.3 mm were used for the experiments to determine the conveying value. The processing parameters were kept constant throughout all the tests, as shown in Table 2. The so-called dwell value $n_v$, which describes the number of revolutions the tool remains in the stop phase, was varied in the steps: 1/1.5/2/2.5/3. This is realized via the adjustment of short-stroke distance $H$ of the drive system in reference to Equation (2). The feed was kept constant to obtain the same cutting conditions (cross-section of the undeformed chip) for all tests. As a result, the periodical width $DP_{theor}$, as shown in Table 3, as well as the conveying cross-section increase with a rising dwell value. Thus, the objective was not to produce geometrically equivalent surface structures but to investigate the influence of the dwell value under the same cutting conditions.

**Table 2.** Constant machining parameters.

| Parameter | Value | Machine System |
|:---:|:---:|:---:|
| Feed $f$ | 0.1 mm | |
| Rotational speed $n$ | 597 min$^{-1}$ | |
| Cutting speed $v_c$ | 150 m/min | SPINNER PD 32 |
| Depth of cut $a_P$ | 50 μm | |
| Back feed velocity $v_{Hub,r}$ | 59.7 mm/min | AeroLas System |

**Table 3.** Variable machining parameters: dwell value, the resulting setting value for the short-stroke distance and the periodical width of the surface structure.

| Dwell Value $n_v$ | Short-Stroke Distance $H$ | Periodical Width $DP_{theor}$ |
|---|---|---|
| 1 | 100 μm | 104 μm |
| 1.5 | 150 μm | 154 μm |
| 2 | 200 μm | 204 μm |
| 2.5 | 250 μm | 255 μm |
| 3 | 300 μm | 305 μm |

In order to assess the influence of the resulting surface topography on the performance in a sealing system with radial shaft seals, the conveying value of the specimens has been determined. The conveying value characterizes the axial conveying effect the surface topography imposes on the lubricant in the gap between the seal lip and the specimen during shaft rotation. Typically, the axial length of this gap (contact width) is in the range of 100 μm to 150 μm, and it can increase to values > 300 μm due to wear. Therefore, from each material type, three types of shaft surfaces have been selected for the test with a periodical width $DP_{theor}$ of about 100 μm, 150 μm, and 300 μm. The conveying value has been determined as the average of two measurements on two different axial positions on the same specimen.

The determination of the conveying value is based on the converse installation of the seal and the two-chamber principle: the seal and the specimen are mounted between two oil chambers, where the air-side chamber is fully flooded, and the oil side chamber represents the conditions in application. In this case, the oil level is in the middle of the horizontal shaft. The rate of fluid pumping from air to oil side can be determined using a flooded riser pipe and a sensor measuring the decrease in the oil fill height-dependent hydrodynamic pressure in the air-side chamber. The resulting value can then be normalized with respect to the sliding distance, yielding the conveying value of the system consisting of shaft surface and seal. The unit of this normalized measurement is therefore μL/m. By measuring the system conveying value for both rotational directions of the shaft, the conveying effect of the shaft and the seal can be separated. More details about the test setup, measurement, and evaluation are given in [17–19].

## 3. Results and Discussion

### 3.1. Surface Analysis

All surfaces created by SST are shown side by side in Figure 5 to ensure visual comparability. The color scaling of the rows with the same dwell value was standardized. In addition, as mentioned, the scaling is inverted as the highest point (the edges of the feedmarks) are at level zero, and the minimum represents the valleys of the groove. In that illustration, the different depth of the grooves resulting from the machining is clearly visible since the edge height is constant due to the kinematical roughness. This visualization style was continued consistently for all surface illustrations. It could be shown that the most significant differences in height occur in hardened steel, while the smallest are found in brass. Furthermore, the height scales increase with increasing dwell value, which can be explained by the higher kinematic roughness due to a larger groove width.

The illustrations in Figure 5 strongly support the hypothesis that the highest material squeezing occurs, in particular, during the SST process of hardened steel, resulting from the initial tool pushing-off and material spring-back effects. In connection with the effective negative rake angle resulting from the chamfer, the tools are deflected into the direction of the passive force during hard machining. Elastic material recovery can also be assumed. As a result, the target depth of the cut is not reached during the first pass. If an additional pass is made, the tool faces the remaining material with a significantly smaller thickness in relation to the cutting-edge radius. In this case, it is possible that the thickness is below the minimum undeformed chip thickness, which does not completely result in a material separation, but in the so-called "ploughing effect"—the material that comes into contact

with the cutting edge is squeezed. Due to this "ploughing effect", the final depth of cut is higher in comparison to the machining of brass, where such effects do not occur because of the significantly smaller tool cutting-edge radius. Due to the rising cross-section of the undeformed chip with increasing dwell value, this effect still occurs for a dwell value of 2.5. In addition, the ploughing effect is less dominant in the machining of the unhardened steel specimen, as the spring-back effect seems to be strongly reduced due to lower process forces. In combination with the largest cutting-edge radius, the plastic deformation of the second pass is the main mechanism. Thus, the depth difference between the two different areas of the groove is not as high as compared the hardened steel. While for a dwell value of 1.5, a variation of the structure depth in the circumferential direction is still visible every half turn, this effect can hardly be observed for a dwell value of 2.5 for soft steel. Furthermore, the alternating change of the groove depth at a dwell value of 1.5 in the cross-section is remarkable (Figure 6, $n_v = 1.5$). For a more accurate analysis, the sections of $n_v = 1$ and $n_v = 2$ were modified, i.e., stretched or stretched along the *x*-axis, and then superimposed. The illustration in Figure 6 shows a high degree of agreement of the relevant range of this superimposition compared with the genuine cross-section of $n_v = 1.5$ in the middle.

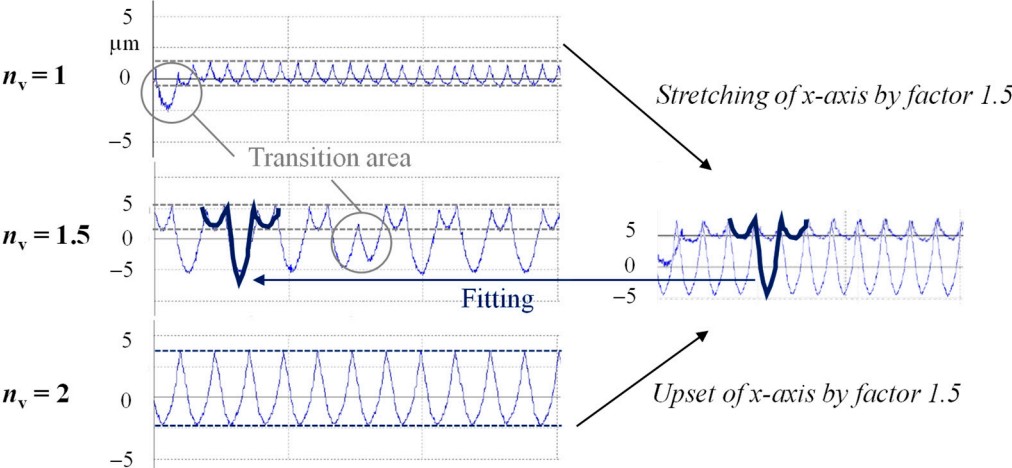

**Figure 6.** Roughness profiles with dwell values of $n_v = 1/1.5/2$ for hardened steel.

There is an increase in the groove depth that does not occur for the kinematical simulations. Additionally, there is no obvious change in the dynamic behavior of the drive system during machining. Its kinematics is characterized by an alternating movement with a fast acceleration during the start phase and a backward motion during the stop phase. The resulting short-stroke path can be measured by a position sensor, and it is depicted in Figure 7 for machining of brass, as-delivered unhardened (A), and hardened steel (H).

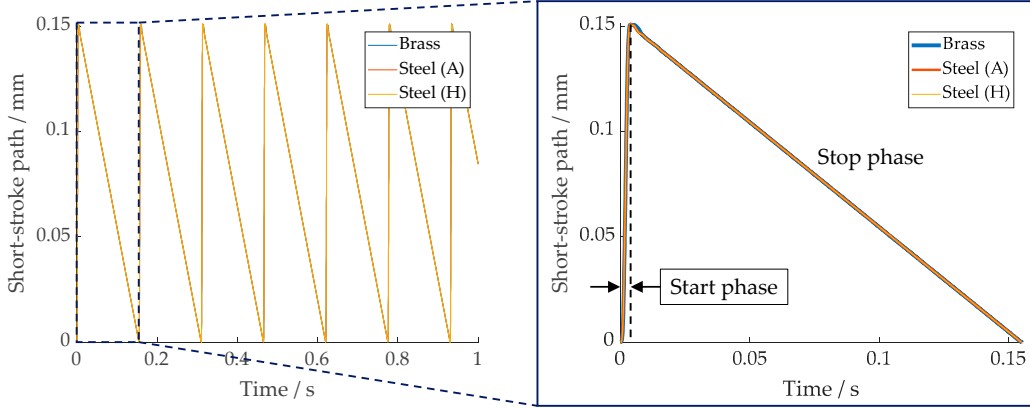

**Figure 7.** Alternating movement of the tool drive system during machining of brass, soft annealed steel (A), and hardened steel (H) with a dwell value of $n_v = 1.5$.

The measured short-stroke path allows for a clear separation of the start and the stop phase. When comparing the different workpiece materials, almost no deviations can be seen in the diagram as the drive system controller can adjust the current position very quickly and precisely to the required target position. Consequently, the alternation of the groove depth does not originate from the machine kinematics and thus has to be a result of the combination of the specimen material and the tool geometry during machining.

Qualitative surface comparison of the transition areas is given in Figure 8 by 3D surface details. In connection with different dimensions of cutting-edge geometry, such as radius or roughness of the cutting edge, they are differently shaped. However, in all cases, the tool pushing-off in the transition area becomes clear, as there is a visible difference in height. In comparison, such differences in height do not occur for the kinematic simulation in Figure 4. In addition, the shape of the transition area, especially for hardened steel (a), supports the hypothesis that plastic material deformation occurs since the exit side of the transition area is interrupted by a sharp edge, which indicates that the material was subsequently deformed there. As a result, the transition area is partially closed, which could hinder the conveying effect of sealed lubricant.

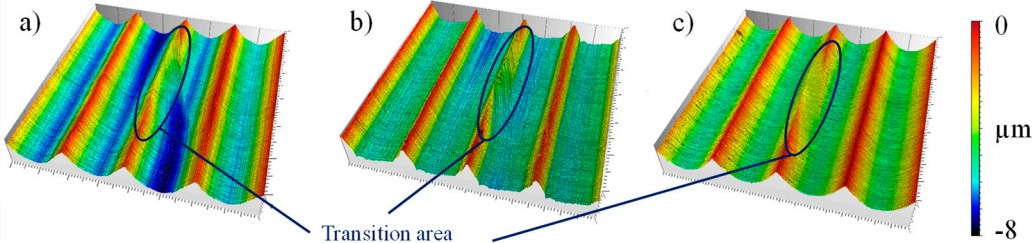

**Figure 8.** Comparison of the shape of the transition areas of hardened (**a**) and unhardened (**b**) steel and brass (**c**) with a dwell value of $n_v = 2$; each field is 800 μm in width and 6° in length.

A further noticeable effect is that the micro-roughness at the unhardened steel specimen is significantly higher due to the comparatively high roughness of the cutting edge, at least for $n_v = 1/1.5/2$. This is particularly evident in the transition area in Figure 8b. For the higher dwell values, a "smoothing" with multiple passes can be observed.

For quantitative comparative purposes, selected parameters are shown in Tables 4 and 5. Here, the 2D roughness parameters square mean roughness $Rq$ and the surface roughness depth $Rz$ are chosen to characterize the profile of the surfaces in the direction of the feed motion because those values are commonly used in industrial production, especially by machining rotary components. They were each averaged from three individually measured values in order to characterize the surface roughness taking into account the different dwell values. Furthermore, the conveying values are represented only for selected specimens due to the high experimental effort. They are determined by an average value from two measured values. $DF$ is the theoretical conveying cross-section that characterizes the cross-section area in the axial direction between two successive roughness peaks. It is dependent on the periodical width and the roughness height, and therefore, $DF$ rises with increasing dwell value.

The periodical width $DP_{MBN}$ calculated according to MBN corresponds for most specimens to the theoretically calculated periodical width $DP_{theor}$. For non-integer dwell values, it is often not possible to place the measuring field in such a way that an influence of the transition areas on the measured values can be excluded. Therefore, these potentially erroneous values are marked with an * in Table 4.

The determination, according to MBN, generates a pseudo-topography from 72 cross-section profiles with an angular distance of 0.5°, which serves as a base for the calculation of the twist surface. Figure 9 compares these calculated surfaces for the reference specimen with twist (above, R) and the hardened steel specimen without twist for a dwell value of 1 (below, H1).

**Table 4.** Overview of the experiments and the results of surface characterization (* affected by the transition area or surface structure).

| | $n_v$ | Periodical Width $DP_{MBN}$ (μm) | Conveying Cross-Section $DF_{MBN}$ (μm²) | $Rq$ (μm) | $Rz$ (μm) | Conveying Value (μL/m) |
|---|---|---|---|---|---|---|
| Reference (R) | - | 0.1 | 92.0 | 0.52 | 2.38 | −0.0017 |
| Hardened (H) 1 | 1 | 0.104 | 84.9 | 0.46 | 2.28 | −0.0015 |
| H 2 | 1.5 | 0.310 * | 588.0 * | 0.70 | 5.44 | −0.0019 |
| H 3 | 2 | 0.205 | 658.0 | 1.39 | 6.21 | - |
| H 4 | 2.5 | 0.253 * | 926.0 * | 2.41 | 8.26 | - |
| H 5 | 3 | 0.308 | 1924.0 | 3.57 | 12.06 | −0.002 |
| Unhardened (A) 1 | 1 | 0.103 | 83.1 | 0.51 | 2.17 | - |
| A 2 | 1.5 | 0.308 * | 304.0 * | 1.54 | 3.67 | - |
| A 3 | 2 | 0.205 | 658.0 | 1.82 | 5.41 | - |
| A 4 | 2.5 | 0.255 | 1308.0 | 2.17 | 8.76 | - |
| A 5 | 3 | 0.305 | 2050.0 | 3.58 | 12.28 | - |
| Brass (B) 1 | 1 | 0.104 | 70.2 | 0.39 | 1.69 | −0.0002 |
| B 2 | 1.5 | 0.154 | 239.0 | 0.86 | 3.25 | −0.0065 |
| B 3 | 2 | 0.205 | 546.0 | 1.49 | 5.23 | - |
| B 4 | 2.5 | 0.255 | 1021.0 | 2.23 | 7.75 | - |
| B 5 | 3 | 0.305 | 1658.0 | 3.10 | 10.63 | −0.0006 |

**Table 5.** Overview of the simulations and the results of surface characterization.

| | $n_v$ | Periodical Width $DP_{theor}$ (μm) | Conveying Cross-Section $DF_{theor}$ (μm²) | $Rq$ (μm) | $Rz$ (μm) | Conveying Value (μL/m) |
|---|---|---|---|---|---|---|
| Simulation (S) 1 | - | 0.1 | 100.7 | 0.46 | 1.52 | - |
| S 2 | 1 | 0.104 | 110.2 | 0.48 | 1.61 | - |
| S 3 | 1.5 | 0.154 | 370.5 | 1.09 | 3.63 | - |
| S 4 | 2 | 0.204 | 808.8 | 1.86 | 6.08 | - |
| S 5 | 2.5 | 0.255 | 1413.1 | 2.69 | 8.72 | - |

Reference surface: twist lead angle $D_\gamma = 0.0228$

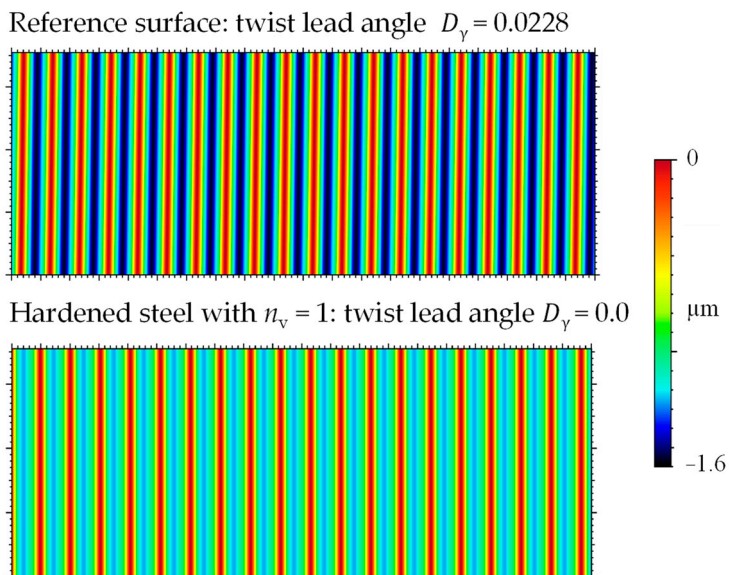

Hardened steel with $n_v = 1$: twist lead angle $D_\gamma = 0.0$

**Figure 9.** Optical comparison of the twist analyses according to MBN, taking the twist lead angle into account; each field is 2 mm in width and 36° in length.

In the calculation of the surfaces by the algorithm according to MBN, the roughness edges are slightly rounded and therefore they are not included in the calculation of the twist parameters, such as the conveying cross-section $DF_{MBN}$. In contrast, the conveying

cross-section calculation within the kinematic simulation ($DF_\text{theor}$ in Table 5) considers the complete kinematic roughness profile, including the roughness edges, resulting in larger conveying cross-sections compared to the measured surfaces evaluated according to MBN. This relationship is shown in Figure 10.

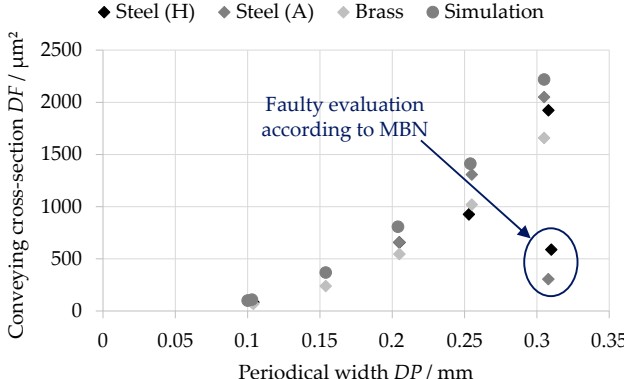

**Figure 10.** Relationship between periodical width and conveying cross-section of the experimental as well as the simulated data.

Figure 10 shows that there is a quadratic relationship between the conveying cross-section and the periodical width. However, the significant differences in the surface structure generated in steel with a non-integer dwell value result in a faulty calculation of the cross-section by the MBN. Furthermore, the deviations between the simulated and measured surfaces increase with rising groove distance.

As the conveying cross-section increases, the oil volume within the surface valleys should also increase. For surfaces with a twist, an increase in the conveying cross-section thus leads to an increase in the conveying volume, as the sealing lip cannot fully be in contact with the specimen in the circumferential direction. According to MBN, the percentage contact length for the twisted reference specimen is 19.3%, which allows the oil to be conveyed underneath the sealing lip. On the other hand, the surfaces without twist have a contact length of 100%, which theoretically corresponds to a parallel alignment of the twist structure to the sealing lip in the circumferential direction and makes it difficult to convey oil out of the working chamber. Since the sealing lip width (100 µm–150 µm in new condition) is, for the most part, less than or equal to the periodical width, an increase in the conveying cross-section for surfaces without twist does not necessarily have to result in an increase in the oil volume in the contact area between the shaft and the sealing lip.

The diagram in Figure 11 shows the average roughness values of all tests, whereby the *Rq* and *Rz* values for the reference specimen are shown as a dashed line across the entire diagram width for better visual comparability. It is clear that with a dwell value of 1, the surface roughness of the twist-free and twist-affected surfaces are almost identical. With an increase in the dwell value, the roughness values increase approximately linearly, which is illustrated by the fitted trend lines.

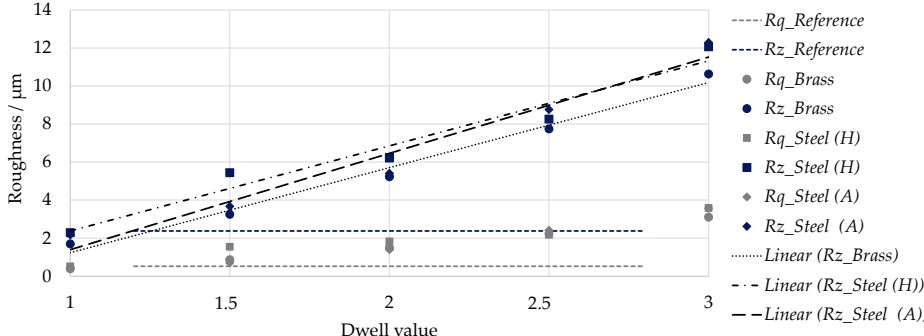

**Figure 11.** Results of the roughness measurement.

### 3.2. Functional Analysis

Due to the low hardness of the brass specimen, extreme shaft wear was observed in the sealing contact after the conveying value measurements, which were conducted until the riser pipe was emptied in each rotational direction (about 10 h, depending on the conveying value of the system). The pressure gradient (proportional to the conveying value) also changed drastically during the measurement time since the wear process removed the original surface topography. Therefore, the tests on the brass specimens were repeated with a different position of the sealing lip on the specimen and a reduced measurement time of about 1 h in each rotational direction in order to reduce the amount of wear during the test. The results depicted in Table 4 are based on this second measurement.

In comparison to the reference value, it is evident in the diagram in Figure 12 that processing by SST has no significant influence on the conveying value for the hardened steel specimens. The conveying values of the brass specimens with the integer dwell values (1 and 2) are clearly below the reference, while those of the specimen with the non-integer (1.5) dwell value have a significantly higher conveying value in amount. However, due to the heavy wear, these values cannot be considered robust enough to establish a well-founded correlation between the conveying value and the dwell value or the structural characteristics of brass, respectively.

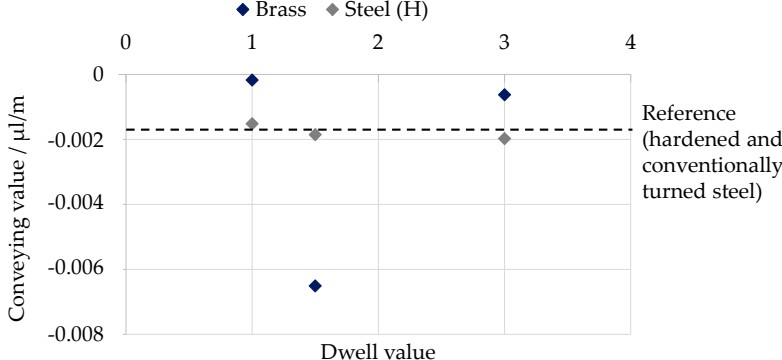

**Figure 12.** Relationship between dwell value and conveying value of selected hardened steel and brass specimens.

It can be assumed that for twist-free structures and a sealing lip width less than or equal to the periodical width, there is no increase in the oil volume in the radial shaft seal contact. Hence, no significant increase in the conveying value is visible in Figure 12 with increasing periodical width. Despite the irregular structural depth in the circumferential direction of the steel specimens for a dwell value of 1.5 (Figure 5), no change in the conveying value can be observed. A possible reason for that could be a deformation of the radial shaft seal. This leads to the conclusion that the explained effects from elastic–plastic deformation do not have any influence on the conveying value. Nevertheless, SST can represent a more productive process in comparison to conventional turning, especially at higher feeds. For brass, a significantly larger conveying value (in amount) was observed for a dwell value of 1.5, which might be a result of the increased wear of the twist structure.

### 4. Summary and Conclusions

The investigations show that the unique kinematic of the SST process results in strongly different twist-free surface structures in reference to the chosen dwell value and the workpiece material. The following conclusions can be drawn from the presented research results:

(1) The differences in comparison to the ideal kinematic surface simulations are a result of the elastic–plastic deformation effects that are dominant in the machining of hardened steel. For additional passes of the tool, the remaining thickness of the material is below the minimum undeformed chip thickness that leads to a ploughing effect.

(2) The ploughing effect is less dominant in the machining of the unhardened steel as the spring back effect seems to be strongly reduced due to lower process forces.

(3) For brass, such effects do not occur because of its low material hardness and the significantly smaller tool cutting-edge radius.

(4) The selection of an integer dwell value is recommended in order to achieve homogeneous twist-free surface structures without a depth difference in the circumferential direction. Additionally, a dwell value > 1 is suggested as a "smoothing" of the surface occurs for multiple passes, and the transition areas can be partially closed by the deformation of material during the SST of steel.

(5) The conveying value is independent of the periodical width $DP$ for twist-free surfaces with periodical widths greater than or equal to the sealing lip width (100 μm–150 μm).

(6) No reduction of the conveying value is achieved for the hardened steel specimens in comparison to the reference specimen with a twist structure.

(7) The comparative investigations with a material exhibiting good machinability, such as brass, are only able to provide limited support, as, in particular, the determination of the conveying value is not robust enough, and, consequently, the meaningfulness of these results is not assured.

(8) The SST method can enhance the economic efficiency of the machining of hardened steel by increasing the feed without affecting the conveying value. Additionally, it offers the potential to accomplish the finishing of, e.g., gear or cam shafts in one machining operation. Consequently, valuable machining time and costs can be saved by the application of this method.

As there are no comparable studies in this field of twist-free turning, the experimental results offer valuable information regarding the machining behavior. These findings provide the foundation for further investigations to optimize the process in terms of its industrial applicability.

## 5. Outlook

Looking ahead, further experimental investigations are necessary to be able to distinguish between the effective separation and deformation processes in order to influence the elastic–plastic material deformation in relation to the workpiece material, and its case hardened state. In this context, a variation of the machining parameters ($v_c$, $a_p$, $f$) and of the tool, geometry is provided. In particular, the experiments should focus on the closing of the transition areas by the targeted deformation of the surrounding material. That could reduce the leakage of lubricant in the sealing system.

Additional investigations should consider the relationship between the dwell value and the machine tool feed. Within the scope of the paper, the periodical width $DP$ of the surface structures increased with rising dwell value. However, according to Equation (4), $DP$ is also dependent on the feed. Hence, future experiments should examine the machining behavior for a constant periodical width with increasing dwell value by adjusting the feed. Smaller periodical widths would result in lower roughness values and, therefore, might enable a reduction of the conveying effect in comparison to conventional turned surfaces.

**Author Contributions:** Conceptualization, R.B. and T.J.; methodology, R.B. and T.J.; software, T.J.; validation, R.B., T.J. and T.S.; formal analysis, T.J.; investigation, R.B., T.J. and T.S.; resources, A.S., O.K. and S.T.; data curation, R.B. and T.J.; writing—original draft preparation, R.B., T.J. and S.T.; writing—review and editing, A.S., S.T. and O.K.; visualization, R.B. and T.J.; supervision, A.S., S.T. and O.K.; project administration, A.S. and S.T.; funding acquisition, A.S. and S.T. All authors have read and agreed to the published version of the manuscript.

**Funding:** This research was funded by the German Research Foundation (Deutsche Forschungsgemeinschaft (DFG, grant number 430170022).

**Institutional Review Board Statement:** Not applicable.

**Informed Consent Statement:** Not applicable.

**Acknowledgments:** The authors gratefully acknowledge the funding of this research by the German Research Foundation (Deutsche Forschungsgemeinschaft, DFG) within the project "Design and manufacturing of radial shafts seals mating surfaces using manufacturing processes with geometrically defined cutting edges".

**Conflicts of Interest:** The authors declare no conflict of interest. The funders had no role in the design of the study; in the collection, analyses, or interpretation of data; in the writing of the manuscript, or in the decision to publish the results.

## Nomenclature

| | | |
|---|---|---|
| $a_\mathrm{p}$ | Depth of cut | µm |
| **CBN** | Cubic boron nitride | |
| **CCC** | Coated cemented carbide | |
| $DP_\mathrm{MBN}$ | Periodical width (according to MBN) | µm |
| $DP_\mathrm{theor}$ | Periodical width (calculated) | µm |
| $DF_\mathrm{MBN}$ | Conveying cross-section (according to MBN) | µm$^2$ |
| $DF_\mathrm{theor}$ | Conveying cross-section (calculated) | µm$^2$ |
| $D_\gamma$ | Twist lead angle | 1° |
| $f$ | Machine tool feed | mm |
| $H$ | Short-stroke distance | mm |
| **MBN** | Mercedes Benz Norm 31007-7 | |
| $n$ | Rotational speed of the specimen | min$^{-1}$ |
| $n_v$ | Dwell value—number of workpiece revolutions in the stop phase | - |
| $r_\beta$ | Cutting-edge radius | µm |
| $r_\varepsilon$ | Corner radius | mm |
| $Rq$ | Square mean roughness | µm |
| $Rz$ | Surface roughness depth | µm |
| $Rz_\mathrm{kin}$ | Kinematical surface roughness depth | µm |
| $Rz_0$ | Background noise from the ambient conditions | µm |
| **SCD** | Single crystalline diamond | |
| **SST** | Start-stop turning | |
| $t_\mathrm{b}$ | Time spent in the start phase | s |
| $t_\mathrm{v}$ | Time spent in the stop phase | s |
| $v_c$ | Cutting speed | m/min |
| $v_\mathrm{f}$ | Feed velocity of the machine tool axis | m/min |
| $v_\mathrm{f,SS}$ | Feed velocity of the tool relative to the workpiece | m/min |
| $v_\mathrm{Hub,r}$ | Short-stroke velocity in the negative direction of the machine tool feed motion/Back feed velocity | mm/min |
| $v_\mathrm{Hub,v}$ | Short-stroke velocity in the positive direction of the machine tool feed motion | m/min |
| $v_\mathrm{res}$ | Velocity of the tool in the start phase relative to the workpiece resulting from the superposition of $v_c$ and $v_\mathrm{f,SS}$ | m/min |
| $\alpha$ | Clearance angle | 1° |
| $\alpha_\mathrm{L}$ | Transition angle of the tool-movement during the startphase | 1° |
| $\beta$ | Wedge angle | 1° |
| $\Delta x$ | Geometrical extension of the intersecting tool body in the direction of the $x$-axis | mm |
| $\Delta \mathrm{y}$ | Geometrical extension of the intersecting tool body in the direction of the $y$-axis | mm |
| **ls** | Gaussian profile filter (short-wave cut-off) | µm |
| $\varphi$ | Inclination angle of the tool body in the $x$-$y$ plane | 1° |

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
