# Peer review of "Elastic–Plastic Material Deformation and Conveying Value of Twist-Free Turned Surfaces"

_surfaces, doi:10.3390/surfaces5030029_

Round 1

Reviewer 1 Report

The authors have used an approach for producing surfaces without twisting, based on new turning kinematics, respectively for 16MnCr5 steel (as received and heat treated) and brass, and analyzed the change of cutting conditions and its effect on the resulting surface structure. I find this problem interesting for practice. The article is structured and written well. Therefore I recommend acceptance after minor correction.

Please add a list of used abbreviations to the nomenclature.

Author Response

Dear reviewer,

Thank you for your valuable comments, points of discussion, and suggestions for improving the manuscript. Our answers or changes are listed below as reactions to your points.

Point 1: Please add a list of used abbreviations to the nomenclature.

Response 1: Abbreviations have been added to the nomenclature.

Sincerely,

The team of authors

Reviewer 2 Report

The Elastic-plastic material deformation and conveying value of twist-free turned surfaces is analysed in this work. The following suggestions are to carried out in the final file

1. Avoid using the university name in the Abstract. 

2. Why do you recommend integer value > 1 for achieving homogeneous surface structure.

3. The comparison result for the materials should be mentioned as numerical values in the conclusion section

4. The measurement of Rq, Ra, Rz are to be cleary mentioned in the paper

5. How the kinematical simulation and dynamic behaviour of the drive system is helpful?

6. Optimization of the results are not done. This may be considered.

7. Include the following literatures in the revised manuscript

https://doi.org/10.1177/0954406212466193

https://doi.org/10.1007/s12206-013-0215-7

8. English language should be checked .

Author Response

Dear reviewer,

Thank you for your valuable comments, points of discussion, and suggestions for improving the manuscript. Our answers or changes are listed below as reactions to your points.

Point 1: Avoid using the university name in the Abstract.

Response 1: The reference to the university has been removed from the abstract.

Point 2: Why do you recommend integer value > 1 for achieving homogeneous surface structure.

Response 2: For clarification, the explanation for the recommended dwell value in section 4 is extended.

Point 3: The comparison result for the materials should be mentioned as numerical values in the conclusion section       .

Response 3: It is unclear to the authors which quantitative statements this comment should refer to, since the structural differences are not characterized by a specific parameter. The robustness of the measured conveying values is discussed at the end of section 3 where it becomes apparent that they are not suitable for a quantitative comparison.

Point 4: The measurement of Rq, Ra, Rz are to be cleary mentioned in the paper.

Response 4: The measurement methodology for recording the profile roughness parameters Rq and Rz was added. The parameter Ra is intentionally not dealt within the paper, as Rq is usually used in the field of sealing technology for surface characterization instead of Ra.

Point 5: How the kinematical simulation and dynamic behaviour of the drive system is helpful?

Response 5: The kinematical simulation serves to represent the "ideal state" of the generated microstructure and thus as a basis for comparison with the real generated surfaces. The dynamic behavior serves to confirm that the microstructural deviations do not originate from the machine kinematics.

Point 6: Optimization of the results are not done. This may be considered.

Response 6: Within the scope of the investigations, a further optimization was not possible due to the limited number of specimens and time available. Nevertheless, we have added section 5 to give an outlook on further investigations.

Point 7: Include the following literatures in the revised manuscript:

https://doi.org/10.1177/0954406212466193,

https://doi.org/10.7158/M12-040.2012.10.2,

https://doi.org/10.1007/s12206-013-0215-7 .

Response 7: The named literature is judged to be not relevant in the context of twist-free turning and therefore it is not included.

Point 8: English language should be checked.

Response 8: We have checked the English language and corrected some minor mistakes.

Sincerely,

The team of authors

Reviewer 3 Report

Dear author(s), please find some comments on the manuscript ‘Elastic-plastic material deformation and conveying value of twist-free turned surfaces’, Manuscript ID: surfaces-1841167:

1.      Considering the “Introduction’ section is well written and interesting for a regular reader. However, some issues must be raised that there is a lack of critical review of the current state of knowledge around the presented topic. Even if motivation is well known, it does not respond to the critical review that is lost. Please try to emphasize the novelty by presenting critical issues.

2.      About all of the equations presented in the manuscript, if they are not newly proposed, but previously published, e.g. by author(s) in already published papers, they should be indicated (referenced). The current presentation is unclear and ambiguous.

3.      For a better understanding of the second section, I suggest numbering each of the subsections, like ‘2.1. Start-stop turning process’, ‘2.2. Resulting twist-free surface structure’ and so on. The same for section no 3. In some cases reader is lost.

4.      It was not justified why results in the simulated transition area (Figure 4) were received with minus values (maximum height was equal to 0). The same in other figures, e.g. Figure 5.

5.      Considering measurement of surface there are no words against measurement uncertainty or measurement errors, e.g. noise. Some clarifications must be provided. Please look and try to refer to the issues presented in the following items:

(1) https://doi.org/10.1088/2051-672X/3/3/035004

(2) https://doi.org/10.3390/ma14175096

(3) https://doi.org/10.3390/app7010054

6.      ‘Summary and Conclusions’ section should be improved significantly(!) that in the current form it is extremely difficult to define the novelty of this section. Moreover, please try to divide this section into separate, numbered gaps.

7.      ‘The Outlook’ section should be provided that currently there is no justification of both the current and future studies. This should be derived from presented studies, nevertheless, for a considered manuscript, should be presented separately.

From all of the manuscript, the reader seems to be lost, unfortunately. So, respectively, suggested aspects must be significantly improved in the manuscript considered.

Author Response

Dear reviewer,

Thank you for your valuable comments, points of discussion, and suggestions for improving the manuscript. Our answers or changes are listed below as reactions to your points.

Point 1: Considering the “Introduction’ section is well written and interesting for a regular reader. However, some issues must be raised that there is a lack of critical review of the current state of knowledge around the presented topic. Even if motivation is well known, it does not respond to the critical review that is lost. Please try to emphasize the novelty by presenting critical issues.

Response 1: The state of the art in science and technology in terms of twist-free turning is very limited. Thus, we believe that this area of research has been sufficiently covered with its advantages and disadvantages. As the paper focuses on the start-stop turning process the introduction points out and references the challenges of this unique machining operation (like the dynamical performance and mechanical stability of the drive unit, the occurrence of transition areas, the characterization of SST surfaces and its functional properties). Furthermore, we have added a sentence that justifies the increased investment costs.

Point 2: About all of the equations presented in the manuscript, if they are not newly proposed, but previously published, e.g. by author(s) in already published papers, they should be indicated (referenced). The current presentation is unclear and ambiguous.

Response 2: The reference for the first introduction of the equations for the description of the dynamic behaviour of the start-stop turning process in section 2.1 and 2.2 has been added. The equations in section 2.3 are newly proposed.

Point 3: For a better understanding of the second section, I suggest numbering each of the subsections, like ‘2.1. Start-stop turning process’, ‘2.2. Resulting twist-free surface structure’ and so on. The same for section no 3. In some cases reader is lost.

Response 3: The additional numbering of the subsections has been added.

Point 4: It was not justified why results in the simulated transition area (Figure 4) were received with minus values (maximum height was equal to 0). The same in other figures, e.g. Figure 5.

Response 4: Although Fig. 5 already contained a corresponding explanation for this display variant, this explanation is also added to Fig. 4.

Point 5: Considering measurement of surface there are no words against measurement uncertainty or measurement errors, e.g. noise. Some clarifications must be provided. Please look and try to refer to the issues presented in the following items:

(1) https://doi.org/10.1088/2051-672X/3/3/035004

(2) https://doi.org/10.3390/ma14175096

(3) https://doi.org/10.3390/app7010054 .

Response 5: From the authors' point of view, there is no reason to refer to this literature. These papers deal in particular with 3D surface evaluation, which is deliberately omitted in the investigations presented. In the case of periodic structures, profile-based roughness parameters are sufficient; in addition, quantitative evaluations based on the MBN and qualitative evaluations of the different characteristics are carried out. The use of areal surface parameters was examined and considered not to be appropriate for the presented facts.

Point 6: ‘Summary and Conclusions’ section should be improved significantly(!) that in the current form it is extremely difficult to define the novelty of this section. Moreover, please try to divide this section into separate, numbered gaps.

Response 6: We have revised the ‘Summary and Conclusions’ section. As there are no comparable studies in this field of twist-free turning the experimental results offer valuable information regarding the machining behaviour during start-stop turning. As mentioned, the start-stop turning offers a higher economic efficiency as the conveying value does not increase with higher feeds in comparison to conventional turning. Nevertheless, we have added section 5 to give an outlook for further investigations to optimize the process in terms of its industrial applicability.

We have used paragraphs to separate the conclusions from each other. Nevertheless, we do not consider a separate numbering, since the results are interdependent and cannot be considered separately.

Point 7: ‘The Outlook’ section should be provided that currently there is no justification of both the current and future studies. This should be derived from presented studies, nevertheless, for a considered manuscript, should be presented separately.

Response 7: We have added an outlook on further investigations in section 5 that is derived from the results of the presented studies.

Sincerely,

The team of authors

Round 2

Reviewer 1 Report

The authors have made the necessary corrections. Therefore, I recommend  the article to be accepted for publication.

Author Response

Dear reviewer,

thank you very much for your positive assessment. 

Sincerely,

The team of authors

Reviewer 2 Report

Accept

Author Response

(The authors gave the same response as above.)

Reviewer 3 Report

Dear authors, a manuscript ‘Elastic-plastic material deformation and conveying value of twist-free turned surfaces’, Manuscript ID: surfaces-1841167, even revised, still have some weakness that makes further evaluation complicated.

Firstly, many responses were addressed flatty, e.g. first (Introduction with the critical review was not improved), fourth (where is the explanation for Figure 4 and 5?), sixth and seventh with a conclusion and outlook performance were not also improved significantly; from both subsections, the novelty is still difficult to be clearly indicated.

Moreover, I found responses for reviewer #2 also provided with not fully performance, e.g. comments no. 3, 4 (parameters were not fully justified) and 6.

From all of your additional (revised) data presented, there are still no satisfactory responses to all of the issues (for each of two from three reviewers) raised according to the primary manuscript.

Most of the responses were addressed too superficially.

Nevertheless, the nightmare comes with the conclusion section that, in my opinion, was not improved significantly, as required and strongly mentioned to the author(s). After reading step by step three times, the conclusions and the outlook, are still far from good understanding for a regular reader.

Author Response

Dear reviewer,

Thank you again for your comments and points of discussion. Our answers or changes are listed below as reactions to your points. In order to ensure consistency, the points and responses from the second  review round have been numbered to avoid duplication. Comments that address one of the points from the first review round again have been marked with "_2". Also, please note that the first revised version of the manuscript incorrectly waived the “Track change” function. We apologize for any inconvenience this may have caused in the tracking of changes. In the resubmitted manuscript, we have a) activated the “Track change” function compared to the previous version, and b) added color markings to indicate the relevant changed passages compared to the first submitted version. We hope this reflects our motivation to address the relevant comments of the reviewers.

Sincerely,

The team of authors

___ 

Response to Reviewer 3 Comments, Round 1

Point 1: Considering the “Introduction’ section is well written and interesting for a regular reader. However, some issues must be raised that there is a lack of critical review of the current state of knowledge around the presented topic. Even if motivation is well known, it does not respond to the critical review that is lost. Please try to emphasize the novelty by presenting critical issues.

Response 1: The state of the art in science and technology in terms of twist-free turning is very limited. Thus, we believe that this area of research has been sufficiently covered with its advantages and disadvantages. As the paper focuses on the start-stop turning process the introduction points out and references the challenges of this unique machining operation (like the dynamical performance and mechanical stability of the drive unit, the occurrence of transition areas, the characterization of SST surfaces and its functional properties). Furthermore, we have added a sentence that justifies the increased investment costs.

Point 2: About all of the equations presented in the manuscript, if they are not newly proposed, but previously published, e.g. by author(s) in already published papers, they should be indicated (referenced). The current presentation is unclear and ambiguous.

Response 2: The reference for the first introduction of the equations for the description of the dynamic behaviour of the start-stop turning process in section 2.1 and 2.2 has been added. The equations in section 2.3 are newly proposed.

Point 3: For a better understanding of the second section, I suggest numbering each of the subsections, like ‘2.1. Start-stop turning process’, ‘2.2. Resulting twist-free surface structure’ and so on. The same for section no 3. In some cases reader is lost.

Response 3: The additional numbering of the subsections has been added.

Point 4: It was not justified why results in the simulated transition area (Figure 4) were received with minus values (maximum height was equal to 0). The same in other figures, e.g. Figure 5.

Response 4: Although Fig. 5 already contained a corresponding explanation for this display variant, this explanation is also added to Fig. 4.

Response 4_2: For a better understanding of the reason for the inverted scales, a comparison of surfaces with “normal” scaling is illustrated below. It is apparent, that this kind of representation is not suitable for focusing on the differences of the material specific cutting behaviour, since there is no reference height level. For this purpose, the scales in the paper are inverted so the peaks of the profile serve as reference and the different valley depths become clearly visible.

(--> please have a look on the attached file)

Point 5: Considering measurement of surface there are no words against measurement uncertainty or measurement errors, e.g. noise. Some clarifications must be provided. Please look and try to refer to the issues presented in the following items:

(1) https://doi.org/10.1088/2051-672X/3/3/035004

(2) https://doi.org/10.3390/ma14175096

(3) https://doi.org/10.3390/app7010054 .

Response 5: From the authors' point of view, there is no reason to refer to this literature. These papers deal in particular with 3D surface evaluation, which is deliberately omitted in the investigations presented. In the case of periodic structures, profile-based roughness parameters are sufficient; in addition, quantitative evaluations based on the MBN and qualitative evaluations of the different characteristics are carried out. The use of areal surface parameters was examined and considered not to be appropriate for the presented facts.

Response 5_2: It is mentioned that all the specimen surfaces shown were recorded tactilely in our investigations. In this respect, the justified comments on measurement uncertainties with optical methods regarding the proposed paper (3) are not relevant here. However, additional phrases have been included in the paper to clarify the origin of the respective measurement data, as well as to provide more detailed information on the measurement methodology used, for example with regard to the used stylus, selected profile distances, applied filters as well as measurement noise.

Point 6: ‘Summary and Conclusions’ section should be improved significantly(!) that in the current form it is extremely difficult to define the novelty of this section. Moreover, please try to divide this section into separate, numbered gaps.

Response 6: We have revised the ‘Summary and Conclusions’ section. As there are no comparable studies in this field of twist-free turning the experimental results offer valuable information regarding the machining behaviour during start-stop turning. As mentioned, the start-stop turning offers a higher economic efficiency as the conveying value does not increase with higher feeds in comparison to conventional turning. Nevertheless, we have added section 5 to give an outlook for further investigations to optimize the process in terms of its industrial applicability.

We have used paragraphs to separate the conclusions from each other. Nevertheless, we do not consider a separate numbering, since the results are interdependent and cannot be considered separately.

Point 7: ‘The Outlook’ section should be provided that currently there is no justification of both the current and future studies. This should be derived from presented studies, nevertheless, for a considered manuscript, should be presented separately.

Response 7: We have added an outlook on further investigations in section 5 that is derived from the results of the presented studies.

Response to Reviewer 3 Comments, Round 2

Point 8: Firstly, many responses were addressed flatty, e.g. first (Introduction with the critical review was not improved), fourth (where is the explanation for Figure 4 and 5?), sixth and seventh with a conclusion and outlook performance were not also improved significantly; from both subsections, the novelty is still difficult to be clearly indicated.

From all of your additional (revised) data presented, there are still no satisfactory responses to all of the issues (for each of two from three reviewers) raised according to the primary manuscript.

[…] Most of the responses were addressed too superficially.

Response 8: As described at the beginning of this article, it is highly likely that numerous changes were overlooked by the avoidance of the "track changes" function. Therefore, we have marked the changes in comparison to the first version in color. Regarding the state of the art in science and technology, we have tried to make a critical comparison between the previous methods of twist-free turning and the unique method of start-stop turning. Additionally, the novelty and the aim of the investigations has been clarified and highlighted at the end of the introduction. Furthermore, a detailed explanation of the reasons for inverting the color scales, including an example, was provided under "respond 4_2".

As discussed in more detail in "respond 10," the sections 4 and 5 have been revised, restructured and expanded in regard to the comments, for example by numbering the conclusions.

Point 9: Moreover, I found responses for reviewer #2 also provided with not fully performance, e.g. comments no. 3, 4 (parameters were not fully justified) and 6.

Response 9: Regarding the fact that Reviewer 2 has no further comments and proposes for the publication of the manuscript, as well as Reviewer 1, we see no need for further changes in terms of the objections of points 3 and 4 since we have already addressed them. An additional justification of the analyzed roughness parameters was made in the paper.

Point 10: Nevertheless, the nightmare comes with the conclusion section that, in my opinion, was not improved significantly, as required and strongly mentioned to the author(s). After reading step by step three times, the conclusions and the outlook, are still far from good understanding for a regular reader.

Response 10: We have significantly revised both the summary and conclusions and the outlook section. The restructuring and numbering of the conclusions should contribute to a better clarity of the topic. Furthermore, a more detailed description and expansion of the described relationships enhances the comprehensibility and emphasizes the novelty.

Round 3

Reviewer 3 Report

Dear author(s), manuscript ‘Elastic-plastic material deformation and conveying value of twist-free turned surfaces’, Manuscript ID: surfaces-1841167, in its current, secondly revised form, was improved more suitably so, respectively, can be further processed by the quality journal as the Surfaces is.

Now responses were addressed more appropriately, resolving the raised issues and providing more details than was presented in the first revision.

Finally looks like the author(s) took the comments seriously.

From the above matter, the manuscript in its current, secondly revised version makes the manuscript more suitable for publication in a quality journal as the Surfaces is.